# Zinc Supplementation, Inflammation, and Gut Integrity Markers in HIV Infection: A Randomized Placebo-Controlled Trial

**DOI:** 10.3390/nu17101671

**Published:** 2025-05-14

**Authors:** Jhony Baissary, Ziad Koberssy, Qian Wu, Abdus Sattar, Ornina Atieh, Joviane Daher, Kate Ailstock, Morgan Cummings, Danielle Labbato, Nicholas T. Funderburg, Grace A. McComsey

**Affiliations:** 1School of Medicine, Case Western Reserve University, Cleveland, OH 44106, USA; 2Division of Medical Laboratory Science, School of Health and Rehabilitation Sciences, The Ohio State University, Columbus, OH 43210, USA; 3Clinical Research Center, University Hospitals, Cleveland Medical Center, Cleveland, OH 44106, USA

**Keywords:** zinc, inflammation, HIV, gut integrity, monocyte activation, metabolic markers

## Abstract

**Background:** Low levels of zinc are prevalent in patients living with HIV and are associated with higher morbidity. Zinc has major immunomodulatory effects. This study aimed to assess the effect of zinc supplementation on inflammatory and gut integrity markers and on zinc levels among HIV patients with zinc deficiency. **Methods:** This was a double-blind randomized placebo-controlled trial assessing the efficacy and safety of zinc supplementation on inflammation and gut markers in people with HIV (PWH) ≥ 18 years old, on stable antiretroviral therapy (ART) with undetectable HIV-1 viral load, and with zinc levels of ≤0.75 mg/L. Participants were randomized 2:1 to zinc gluconate tablets at a dose of 90 mg of elemental zinc or a matching placebo daily for 24 weeks. At baseline and at week 24, we measured plasma levels of zinc and markers of inflammation and gut barrier integrity. **Results:** Among the 95 participants enrolled in this study, 74% were male, and 65% were non-white, with a median CD4 count of 722 cells/μL. The primary analysis showed an increase in zinc levels in the active group. A decrease in the monocyte activation marker soluble CD14 was observed in the treatment group at −56.31 ng/mL (−263.24; 134.19), compared to an increase in the placebo group of 101.71 ng/mL (−90.50; 243.20); *p* = 0.021. The stratified analysis showed that the group with the lowest zinc levels at baseline had the greatest improvements in soluble CD14 levels during zinc supplementation. No changes were seen in other inflammation markers or gut integrity markers. **Conclusions:** This is the most comprehensive study on the effect of zinc supplementation in PWH on inflammatory and gut integrity markers. Decreases were seen in the monocyte activation marker sCD14. In the contemporary HIV era with potent effective therapies, suppressed viremia, and high CD4 cells, zinc supplementation does not offer consistent benefits on inflammation.

## 1. Introduction

People living with HIV (PLHIV) have a 5 to 10 years shorter life expectancy than people without HIV (PWoH), with malnutrition being a leading factor in increasing the risk of mortality and opportunistic infections [1]. Understanding the relationship between nutrition and HIV is crucial for comprehensive care. Malnutrition can result from the virus itself, its complications, including sarcopenia and enhanced immune activation, opportunistic infections, or antiretroviral therapy, and it is indicative of a poor prognosis and may accelerate progression to Acquired Immune Deficiency Syndrome (AIDS) [2,3]. A complex interplay exists between nutritional status and HIV infection: immune impairment caused by HIV leads to malnutrition, which, in turn, exacerbates immune deficiency.

In PLHIV, zinc is one of the most prevalent micronutrient deficiencies, affecting more than half of PLHIV [4]. In addition, inadequate zinc levels are associated with increased mortality in PLHIV [5]. Zinc, the second most abundant trace element in the human body following iron, is an essential mineral for catalyzing over 100 enzymatic activities, providing proteins structural support, and regulating gene expression [6,7]. It is crucial to anabolic processes like growth and wound healing; hence, its deficiency affects rapid turnover tissues, leading to delayed puberty and hypogonadism, diarrhea, alopecia, glossitis, nail dystrophy, and immune dysfunction [6]. Zinc is not abundantly stored in the body, requiring regular dietary intake. It is primarily found in animal products, including red meats and various seafood [6]. Despite its importance, zinc deficiency is a significant, yet often underestimated, cause of comorbidity worldwide [8]. The widespread deficiency has been linked to the loss of more than 28 million Disability-Adjusted Life Years (DALYs), particularly in resource-limited countries. In these areas, access to animal-based zinc-rich foods is limited, and diets essentially rely on plant-based sources that contain phytates, an inhibitor of zinc absorption, with a phytate-to-zinc molar ratio exceeding 20 in most of these regions [8].

The first system known to be compromised, even with mild zinc deficiency, is the highly proliferative immune system, leading to increased mortality and morbidity from infectious diseases. This deficiency results in T-cell depletion, lymphopenia, and suppression of both humoral and cellular immunity [9]. Zinc supplementation has been associated with anti-inflammatory and antioxidant properties, primarily due to the upregulation of the expression of two anti-inflammatory zinc finger proteins, A20 and PPAR-α, consequently decreasing the activation of Nuclear Factor kappa-light-chain-enhancer of activated B cells (NF-κB) and leading to a reduction in inflammatory cytokine production [10,11,12]. Furthermore, in a pilot exploratory study, we reported that zinc supplementation in PLHIV on antiretroviral therapy (ART) may attenuate systemic inflammation and improve markers of intestinal barrier function [13]. Such a benefit would be significant, as a compromised intestinal barrier contributes to microbial translocation and continuous immune activation and disease progression in PLHIV [14].

Additionally, despite a tremendous improvement in HIV-related mortality and morbidity due to increased accessibility and affordability of ART, PLHIV remain at higher risk for cardiometabolic disorders, including weight gain, dyslipidemia, hypertension, diabetes, insulin resistance, and endothelial dysfunction, compared to the general population [15,16]. Numerous studies have highlighted the importance of zinc supplementation through upregulation of PPAR-α in improving traditional cardiovascular risk factors such as obesity, atherosclerosis, diabetes mellitus, and lipid and glucose metabolism [12,17].

Our study evaluated the efficacy and safety of a 24-week regimen of zinc supplementation in zinc-deficient people living with HIV receiving stable ART. The primary objectives were to evaluate changes in zinc levels after supplementation and the effects of the change on markers of inflammation and immune activation in the participants. Secondary objectives included examining the intervention’s outcomes on metabolic and cardiovascular indices and on gut integrity markers and assessing the safety and tolerability of zinc supplementation in this population.

## 2. Methods

### 2.1. Study Design and Population

This was a 24-week double-blind, randomized, placebo-controlled trial in adult PLHIV on stable ART for at least 12 weeks, with a documented HIV-1 RNA viral load of ≤400 copies/mL in the last 4 months prior to study entry, and with zinc deficiency (zinc levels ≤ 0.75 mg/L). Pregnant or lactating women, patients with other active inflammatory conditions, individuals diagnosed with active neoplastic diseases requiring chemotherapy and/or use of immunosuppressive drugs, uncontrolled diabetes, and known cardiovascular disease were excluded. Additional exclusion criteria included the following: regular use of agents that may affect inflammation in the last 3 months, consuming supplements containing more than the recommended dietary allowance, RDA, level of nutrients known to affect the immune response, body mass index (BMI) < 18 kg/m^2^, allergy or intolerance to zinc sulfate, excessive alcohol or recreational drug use with poor history of compliance, and any of the following laboratory findings: AST/ALT> 2.5 × ULN, hemoglobin < 9.0 g/dL, or Glomerular Filtration Rate < 50 mL/min. This study was conducted between 18 March 2020 and 2 August 2024 at the University Hospitals Cleveland Medical Center in Cleveland (UHCMC), OH, USA.

### 2.2. Ethical Considerations

The UHCMC Institutional Review Board approved the study protocol (STUDY20190915). Signed written informed consent was required before participation. The drug was provided free of charge to enrolled participants for the duration of the study. Trial number (ClinicalTrials.gov): NCT05085834. An independent Data and Safety Monitoring Board (DSMB) reviewed data safety and adverse events in 6-month intervals.

### 2.3. Intervention Details

The study statistician used the SAS software version 9.4, to create a 2:1 randomization schedule with permuted variably sized blocks. This randomization schedule was accessible only to the site research pharmacists, whereas participants, investigators, study staff, and the study statistician were blinded to treatment allocation. Normal serum zinc levels fall normally between 75 and 150 μg/dL, and the recommended dietary allowance (RDA) of zinc is 8 mg/day and 11 mg/day for adult females and males, respectively [6]. The intervention duration was 24 weeks, and 63 patients were randomized to receive 90 mg of zinc gluconate daily, while 32 were randomized to a daily matching placebo.

### 2.4. Study Measurements

#### 2.4.1. Clinical Assessment of Participants

We collected baseline characteristics of patients, including demographic characteristics, physical activity status, and social, personal, and family history of cardiovascular disease (CVD) and diabetes. Additionally, patients underwent targeted physical examinations, including vital signs measurements.

#### 2.4.2. Metabolic and Cardiovascular Biomarkers

All participants underwent anthropometric measurements, including hip and waist circumference, weight, and height. Serum metabolic measurements included levels of triglycerides (TG), total cholesterol, very-low-density lipoprotein (VLDL), low-density lipoprotein (LDL), high-density lipoprotein (HDL), and non-HDL cholesterol. We used the EndoPAT^®^-2000 device (Itamar Medical, Caesarea, Israel) as an indirect and non-invasive tool to assess endothelial function, as we detailed in a previous study, generating a Reactive Hyperemic Index (RHI, normal is >1.67) and an Augmentation Index corrected to a heart rate of 75 beats per minute (AI 75, lower values reflecting better elasticity) [16]. We calculated the 10-year atherosclerotic cardiovascular disease (ASCVD) score at baseline and 24 weeks using the ASCVD Risk Estimator Plus of the American College of Cardiology [18].

#### 2.4.3. Inflammatory and Gut Biomarkers

Participants were instructed to fast for at least 12 h before blood collection. All blood samples were sent to a CLIA-certified laboratory in real time for hematology, chemistry, and metabolic measurements. HIV-1 RNA and CD4 counts were obtained from the clinical chart, as these are a part of routine HIV care. Additional plasma samples were promptly processed and stored at −80 °C within 2 h and shipped in batches without prior thawing to Dr. Funderburg’s laboratory at Ohio State University for the measurement of biomarkers of inflammation, monocyte activation, and gut integrity using enzyme-linked immunosorbent assays (ELISA). Monocyte activation and inflammatory and endothelial biomarkers included soluble CD14 and CD163 (sCD14 and sCD163), high-sensitivity C-reactive protein (hsCRP), interleukin-6 (IL-6), interferon-gamma inducible protein 10 (IP-10), soluble tumor necrosis factor receptors 1 and 2 (sTNF-RI and sTNF-RII), intercellular adhesion molecule (ICAM), and vascular cell adhesion molecule (VCAM) using ELISA kits from R&D Systems (Minneapolis, MN, USA). Levels of D-dimer and oxidized low-density lipoprotein (oxLDL) were measured using ELISA kits from Diagnostica Stago (Parsippany, NJ, USA) and Uppsala (Mercodia, Uppsala, Sweden), respectively. Gut biomarkers that were measured included zonulin (Immundiagnostik AG, Germany) and intestinal fatty-acid-binding protein (I-FABP, R&D Systems), and markers of bacterial and fungal translocation included lipopolysaccharide-binding protein (LBP, R&D Systems) and β-d-glucan (BDG, MBS756415, MyBioSource, San Diego, CA, USA), respectively.

### 2.5. Statistical Analysis

All primary analyses were performed using data collected at baseline and week 24, and safety visits were conducted at weeks 6, 12, and 18 to collect any potential adverse events and to ensure adherence to the study drug. Descriptive statistics were used to summarize the baseline characteristics and biomarker levels of both the zinc-supplemented and control groups. Continuous variables are presented as the median (IQR). Categorical variables are presented as numbers (percentages). Inter-group comparisons were performed using either two-sample *t* tests or Wilcoxon’s rank-sum tests, depending on the data distribution.

To assess the effect of zinc supplementation on the outcome variables, we used generalized estimating equations, adjusted for relevant covariates, including age and sex, and we examined interactions between time and treatment. To assess the true effect size of zinc supplementation on biomarkers, we considered combining linear coefficients in the equation while accounting for the multiplicative effects of treatment and time. Before using the generalized estimating equation model, appropriate log transformations were applied to outcome variables with skewed distributions to adjust them into relatively symmetric normal distributions. For highly skewed outcome variables, we applied median regression for clustered data with bootstrap replication in standard error estimation.

Stratified analyses were also performed according to the median baseline zinc level to examine whether zinc supplementation had different effects on participants with low and high baseline zinc levels. Each stratum was analyzed separately using generalized estimating equation models or median regression to assess the effect of zinc supplementation on the biomarker changes. Linear combinations of regression coefficients were also used to estimate the true effect size.

We used Spearman’s correlational analysis to estimate the association between zinc levels and a targeted set of biomarkers. The significance level for all statistical tests was set at 0.05. All analyses were performed using Stata 18.0.

## 3. Results

### 3.1. Baseline Characteristics

Overall, 266 individuals were screened for this trial, with n = 171 not meeting the inclusion criteria, mostly (n = 162) due to a zinc level above the threshold required for enrollment. A total of 95 participants were enrolled and randomized to either the treatment (n = 63) or the control group (n = 32). Among the participants enrolled, the average age was 52 years, 74% were male, and 63% were non-white (Table 1). At the baseline visit, participants between the two study groups had a similar (*p* > 0.05) distribution of age, sex, race, ethnicity, BMI, smoking, alcohol use, HIV viral load, and absolute CD4^+^ T-cell counts.

At baseline, the zinc level was 69.00 ug/L (61.20–73.00) in the placebo group vs. 70.00 ug/L (66.00–72.20) in the treatment group (*p* = 0.61). Across both treatment arms, 79% were on integrase inhibitors, 15% on protease inhibitors (PI), and 20% on non-nucleoside reverse transcriptase inhibitors (NNRTIs). Baseline inflammatory, gut integrity, and monocyte activation markers were similar between the two groups (*p* > 0.05 for all). Specifically, sCD14 levels were 1727.05 ng/mL (1525.77–1963.90) in the placebo group vs. 1671.20 ng/mL (1424.41–2001.68) in the treatment group (*p* = 0.68) (Table 1).

### 3.2. Safety Analysis and Adverse Events

Over 24 weeks, 76 participants reported a cumulative total of 250 adverse events, AEs (165 adverse events in the treatment group and 85 adverse events in the control group). Among the 95 participants in the study, 19 participants dropped out prematurely, out of whom 11 participants were lost to follow-up, 1 participant dropped out because of their inability to swallow the pill, 1 non-study-related death was reported (intentional drug overdose), and 6 participants dropped out after week 6 or week 12 of treatment due to adverse events (n = 1 had Grade 1 nausea/vomiting, and n = 5 had Grade 1 nausea); all of these 6 participants were in the treatment group (Figure 1). The distribution of the AE severity was as follows: Grade 1 (n = 206), Grade 2 (n = 36), Grade 3 (n = 6), and Grade 4 (n = 2), with the most common AEs being hyperglycemia (19% of the total AEs), elevated creatinine levels (16% of the total AEs), and hypertriglyceridemia (14% of the total AEs) (Appendix A).

### 3.3. Primary Analysis

The zinc levels in the treatment group increased significantly by 33.50 μg/dL (14.00–62.70) compared to 8.60 μg/dL (1.65–20.00) for the control group at week 24 (*p* < 0.01). A significant relative increase in zinc levels among the treated group was also observed, with a 53.17% (19.44–87.08) rise compared to 12.92% (2.36–31.15) for the control group (*p* < 0.0001). The CD4 counts did not change significantly between baseline and week 24 in either group, and all participants remained virologically suppressed at week 24 (Table 2). There were no significant changes in the inflammatory and gut integrity markers in either the placebo or treatment groups; however, the monocyte activation marker sCD14 decreased significantly in the treatment group over the 24 weeks (*p* = 0.02), with a change of −56.31 ng/mL (−263.24; 134.19), while the control group experienced an increase in this marker by 101.71 ng/mL (−90.50; 243.20). A significant relative change for sCD14 (*p* = 0.02) of 6.57% (−5.25; 17.33) in the control group and −2.87% (−15.74; 7.84) in the treatment group was also observed (Table 2). The metabolic and cardiovascular markers did not change significantly in either group (Table 3).

### 3.4. Stratified Analysis

We next stratified the data based on the median zinc level at baseline. Stratification analysis showed that both groups, below and above the median zinc level, exhibited an increase in serum zinc levels; however, the analysis revealed that the group with below-median zinc levels at baseline showed an improvement in sCD14, with a trend toward significance, exhibiting a decrease of −0.12 (*p* = 0.07). On the other hand, the above-median zinc level group showed a decrease in diastolic blood pressure of −5.67 (*p* = 0.06) and an increase in zonulin of +0.31 (*p* = 0.03). Other inflammatory, metabolic, and gut integrity markers showed no significant changes in either the below- or above-median zinc level groups in week 24 (Table 4).

### 3.5. Correlation Analysis

At baseline, the zinc level showed a significant inverse correlation with IFABP and VCAM, with a correlation coefficient of −0.2286 (*p* = 0.03) and −0.2043 (*p* = 0.05), respectively. None of the other markers correlated with the zinc level. At week 24, zinc levels were positively correlated with triglyceride levels and with 10-year cardiovascular disease risk score [0.2715 (*p* = 0.02) and 0.286 (*p* = 0.01), respectively]. The other variables showed no significant correlation with zinc levels in week 24 (Table 5).

## 4. Discussion

In this double-blind, randomized, placebo-controlled clinical trial, we assessed the effects and safety of zinc supplementation in zinc-deficient patients living with HIV and on stable ART. Inflammatory, immune activation, and gut integrity markers were measured and compared between the treatment group and the placebo group while evaluating the changes in metabolic and cardiovascular factors in each group at weeks 0 and 24 of treatment. Although our study did not show a consistent effect of zinc supplementation on inflammation, gut integrity markers, and cardiovascular risk, the study showed that the zinc level significantly increased in the treatment group, and the monocyte activation marker sCD14 decreased significantly after 24 weeks of treatment compared to an increase measured in the placebo group. In this clinical trial, we demonstrated the tolerability and safety of zinc supplementation in PLHIV, except for some gastrointestinal adverse events reported among our participants.

Studies have shown that higher soluble CD14 levels are independently correlated with an increased risk of all-cause mortality in PLHIV. Sandler et al. and Chevalier et al. have demonstrated in their studies that sCD14 could be an indicator of HIV progression [19,20]. Persistently increased monocyte activation, reflected by a higher sCD14 level, is seen in PLHIV even after ART initiation [21]. While we report a modest decrease in sCD14 levels among the treatment group, decreases in monocyte activation may improve end organ disease and morbidity and mortality in PLHIV.

Thus, our study comes as complementary to a pilot clinical trial performed in our research center at University Hospitals, Cleveland Medical Center, Cleveland, Ohio, which was published in November 2020. The previous pilot study compared changes in groups taking different zinc supplementation doses over 16 weeks without a placebo group. That study showed that zinc supplementation increased zinc levels and decreased sCD14 levels, which corresponded to the results of this study [13].

The results of a systematic review and meta-analysis evaluated 13 experimental studies involving 802 PWH in the active zinc group and 742 PWH in the control group align with the results of our study, where zinc levels significantly increased in the zinc supplementation treatment arms of these studies [2]. This systematic review also assessed the changes in CD4^+^ T-cell counts and viral load after zinc treatment and confirmed a positive effect on CD4^+^ T-cell counts without any change in the viral loads of participants. In our study, we did not see an increase in CD4^+^ T-cell counts or viral loads; however, our population had higher baseline CD4 counts, and the design was selected for enrolling only virologically suppressed individuals on ART [2].

A previous 18-month randomized placebo-controlled trial of zinc supplementation in PWV showed a significant increase in zinc levels in the treatment group compared to the placebo group and a greater reduction in morbidity among the treatment group [4]. This study was of an 18-month duration, unlike ours, where the intervention was only 24 weeks. The previous study also lacked an assessment of inflammatory markers, so no insights into the potential role for decreasing inflammation in the reduction of morbidity reported in this study can be gained.

The results of the ZINC trial, a randomized placebo-controlled clinical trial that enrolled 254 PLHIV and with heavy alcohol use, showed no improvement in mortality rates and cardiovascular disease, which aligns with the results of our study, in which neither the 10-year ASCVD scores or endothelial function assessment changed [22]. Further, the ZINC trial also did not find significant improvement in any inflammatory, gut integrity, or monocyte activation marker measured, which is mostly consistent with the results of our study, except for the decrease in sCD14 we report [22].

Our study has several strengths, particularly its randomized placebo-controlled study design and the extensive assessments of systemic inflammation, gut integrity, and metabolic and cardiovascular markers. There are a few limitations that we need to acknowledge. The intervention lasted for 24 weeks, which may be too short to see significant changes in markers of inflammation or gut integrity, despite the rise in zinc levels shown in the study. Additionally, the participants of our study were well controlled on ART, and we did not consider people with uncontrolled disease who might experience greater benefits. Lastly, our study was completed in the United States, so we cannot generalize the results to resource-limited countries. While the majority of markers of inflammation and gut barrier integrity did not change in our study, we did find a significant change in sCD14 levels between the placebo and treatment arms, and levels of this marker have been associated with HIV pathogenesis and end organ disease in other studies.

## 5. Conclusions

This is the first placebo-controlled clinical trial to assess the safety and effectiveness of zinc supplementation for virologically suppressed people with HIV while conducting a comprehensive assessment of inflammatory, cardiovascular, and gut integrity markers. Although most of these markers showed no change, zinc levels increased and soluble CD14 levels decreased during treatment. Further studies, over a longer treatment period, could be helpful to determine the effects of zinc supplementation on inflammation and immune activation in PLHIV.

## Figures and Tables

**Figure 1 nutrients-17-01671-f001:**
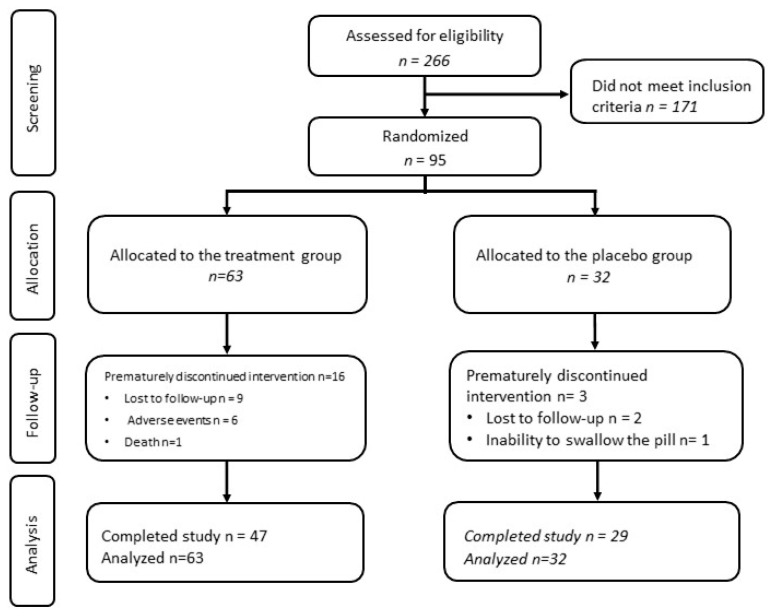
Flow diagram of participants screening, allocation, follow-up, and analysis.

**Table 1 nutrients-17-01671-t001:** Baseline characteristics.

		Both Groups	Placebo	Zinc 90 mg	*p*-Value *
		N = 95	N = 32	N = 63	
	** Age (years) **	52.42 (40.34–60.12)	53.29 (45.31–58.90)	52.42 (38.92–60.55)	0.65
	** Male **	70 (74%)	26 (81%)	44 (70%)	0.23
	** Race **				0.80
	African American	56 (59%)	18 (56%)	38 (60%)	
	Caucasian	37 (39%)	14 (44%)	23 (36%)	
	Native American	1 (1%)	0 (0%)	1 (2%)	
	Other	1 (1%)	0 (0%)	1 (2%)	
	** Ethnicity **				0.38
	Hispanic or Latino	11 (12%)	5 (16%)	6 (10%)	
	**Alcohol** (current)	66 (69%)	21 (66%)	45 (71%)	0.19
	**Smoking** (current)	29 (31%)	13 (41%)	16 (25%)	0.31
** Zinc Level **					
	Zinc (μg/dL)	69.80 (64.30–73.00)	69.00 (61.20–73.00)	70.00 (66.00–72.20)	0.61
** HIV Profile **					
	CD4+ T-cell count	722.00 (526.00–973.00)	668.00 (507.00–775.00)	747.00 (529.00–1024.00)	0.18
	HIV RNA (<20 copies) (%)	74 (80%)	23 (72%)	51 (81%)	0.31
	ART duration (months)	171.86 (108.22–236.36)	167.86 (115.14–239.51)	178.76 (107.20–232.51)	0.88
	PI USE	15 (16%)	7 (23%)	8 (13%)	0.25
** Metabolic Markers **					
	BMI (kg/m^2^)	28.08 (24.62–32.70)	27.45 (23.23–32.40)	28.34 (24.80–32.70)	0.46
	Waist circumference (cm)	97.00 (89.17–106.33)	97.83 (87.50–106.42)	96.67 (89.83–106.33)	0.94
	Systolic blood pressure (mmHg)	126.00 (117.00–141.00)	129.00 (118.50–141.00)	125.00 (117.00–139.00)	0.55
	Diastolic blood pressure (mmHg)	81.00 (75.00–86.00)	82.00 (77.00–88.50)	80.00 (75.00–85.00)	0.30
	Non-HDL cholesterol (mg/dL)	115.30 (95.70–149.20)	115.50 (93.90–166.25)	115.30 (98.00–147.60)	0.79
	HDL (mg/dL)	47.00 (39.90–61.10)	48.15 (41.30–62.70)	45.40 (39.80–60.80)	0.48
	LDL (mg/dL)	96.00 (71.00–118.00)	89.00 (66.50–127.50)	96.00 (74.00–114.00)	0.81
	VLDL (mg/dL)	21.00 (15.00–28.50)	20.00 (15.00–27.00)	22.00 (15.00–29.00)	0.69
	Cholesterol (mg/dL)	169.00 (144.00–202.00)	165.50 (147.50–213.50)	171.00 (144.00–192.00)	0.81
	Chol:HDL ratio	3.40 (2.70–4.40)	3.20 (2.55–4.85)	3.50 (2.80–4.40)	0.47
	Triglycerides (mg/dL)	105.00 (74.00–146.00)	103.00 (76.50–136.00)	108.00 (71.00–146.00)	0.89
	Insulin (uIU/mL)	10.50 (6.00–18.00)	11.50 (6.50–18.50)	10.00 (6.00–17.00)	0.84
	10-year ASCVD (%)	6.50 (2.90–11.80)	6.70 (3.85–12.50)	6.10 (2.70–11.10)	0.44
	Presence of metabolic syndrome	40 (42%)	14 (44%)	26 (41%)	
** Endothelial Function **					
	Reactive Hyperemic Index	1.90 (1.60–2.19)	1.89 (1.62–2.04)	1.91 (1.60–2.21)	0.47
	Augmentation Index	10.00 (−1.00–15.00)	11.00 (4.00–16.00)	7.50 (−2.00–14.00)	0.22
** Inflammatory Markers **					
	hsCRP (ng/mL)	2202.14 (993.51–7795.00)	1998.85 (967.08–6778.08)	2240.00 (1075.40–7930.31)	0.66
	sCD14 (ng/mL)	1681.63 (1468.60–1966.05)	1727.05 (1525.77–1963.90)	1671.20 (1424.41–2001.68)	0.68
	sCD163 (ng/mL)	568.55 (420.62–780.63)	593.61 (461.03–784.74)	543.40 (366.03–778.58)	0.26
	sTNF-RI (pg/mL)	1006.42 (881.12–1195.08)	977.39 (819.30–1200.68)	1036.85 (890.50–1183.57)	0.54
	sTNF-RII (pg/mL)	2234.54 (1907.77–2912.41)	2253.23 (1923.78–3031.33)	2162.27 (1889.96–2867.83)	0.63
	D-dimer (ng/mL)	536.22 (366.79–809.72)	520.93 (293.47–717.11)	558.43 (389.36–883.46)	0.67
	oxLDL (U/L)	50145.48 (39945.47–70630.87)	47606.57 (37342.25–62313.38)	51731.12 (41091.65–75262.57)	0.24
	IL−6 (pg/mL)	1.94 (1.31–3.46)	1.89 (1.42–3.51)	2.11 (1.26–3.46)	0.70
	VCAM (ng/mL)	783.99 (671.49–970.46)	857.01 (693.53–1013.78)	758.04 (658.95–950.49)	0.22
	ICAM (ng/mL)	247.37 (180.09–293.93)	250.13 (175.81–290.17)	244.28 (180.09–298.24)	0.70
	IP−10 (pg/mL)	134.48 (99.61–199.31)	134.70 (90.04–190.78)	130.52 (102.35–207.19)	0.55
** Gut Integrity Markers **					
	LBP (μg/mL)	16.6 (12.77–25.77)	16.44 (12.55–20.35)	16.6 (13.16–30.47)	0.33
	IFAB (pg/mL)	1730.93 (1087.46–2681.82)	1616.94 (996.70–2295.69)	1826.49 (1131.22–2741.16)	0.24
	BDG (pg/mL)	106.08 (77.72–168.20)	122.51 (87.07–197.11)	98.46 (72.89–164.33)	0.24
	Zonulin (μg/mL)	1030 (674.68–1440)	1040 (665.92–1460)	1020 (674.68–1390)	0.80

* For continuous variables, either Wilcoxon’s rank sum tests or two-sample *t* tests were used (as appropriate). For categorical variables, chi-square tests were used. ART = antiretroviral therapy; PI = protease inhibitor; BMI = body mass index; LDL = low-density lipoprotein; HDL = high-density lipoprotein; VLDL = very-low-density lipoprotein; ASCVD: atherosclerotic cardiovascular disease; hsCRP = high-sensitivity C-reactive protein; sCD14 = soluble CD14; sCD163 = soluble CD163; sTNF-RI = soluble tumor necrosis factor receptor−1; sTNF-RII = soluble tumor necrosis factor receptor−2; oxLDL = oxidized LDL; IL−6 = interleukin−6; VCAM: vascular cell adhesion molecule−1; ICAM = intercellular adhesion molecule−1; IP-10 = interferon-gamma-inducible protein of 10 kDa; LBP = lipopolysaccharide-binding protein; IFAB = intestinal fatty-acid-binding protein; BDG = (1,3)-β-d-glucan.

**Table 2 nutrients-17-01671-t002:** Absolute and percent change from baseline to week 24 *.

		Placebo	Zinc 90 mg	*p*-Value
	** Absolute change **			
** Zinc Level **				
	Zinc (μg/dL)	8.60 (1.65–20.00)	33.50 (14.00–62.70)	<0.01
** HIV Profile **				
	CD4+ T-cell count	0.00 (−109.00–73.00)	0.00 (−63.00–38.00)	0.79
** Metabolic Markers **				
	BMI (kg/m^2^)	0.28 (−0.52–1.16)	0.10 (−0.58–0.95)	0.67
	Weight (lbs)	2.20 (−3.50–7.60)	0.00 (−4.00–4.40)	0.42
	Waist-umbilicus (cm)	0.50 (−2.53–2.50)	1.67 (−2.67–3.00)	0.39
	Systolic blood pressure (mmHg)	6.00 (−4.00–12.00)	0.00 (−9.00–9.00)	0.57
	Diastolic blood pressure (mmHg)	3.00 (−5.00–6.00)	0.00 (−5.00–5.00)	0.89
	Non-HDL cholesterol (mg/dL)	3.10 (−28.20–26.50)	1.35 (−12.30–10.85)	0.64
	HDL (mg/dL)	−1.70 (−4.60–1.40)	−2.45 (−8.45–1.25)	0.49
	LDL (mg/dL)	6.00 (−28.00–16.00)	−1.50 (−19.50–11.50)	0.38
	VLDL (mg/dL)	0.50 (−6.00–3.50)	2.00 (−2.00–6.00)	0.23
	Cholesterol (mg/dL)	2.00 (−25.00–27.00)	−3.50 (−23.50–12.00)	0.47
	Chol:HDL ratio	0.10 (−0.20–0.40)	0.10 (−0.10–0.45)	0.52
	Triglycerides (mg/dL)	2.00 (−29.00–12.00)	7.00 (−12.50–34.50)	0.21
	10-year ASCVD (%)	0.40 (−1.10–2.00)	0.00 (−1.35–1.55)	0.61
** Endothelial Function **				
	Reactive Hyperemic Index	−0.34 (−0.77–0.16)	−0.18 (−0.60–0.23)	0.77
	Augmentation Index	0.00 (−8.00–5.50)	−1.00 (−7.00–6.00)	0.48
** Inflammatory Markers **				
	hsCRP (ng/mL)	−22.86 (−689.27–1744.48)	1.33 (−602.56–1174.87)	0.97
	sCD14 (ng/mL)	101.71 (−90.50–243.20)	−56.31 (−263.24–134.19)	0.02
	sCD163 (ng/mL)	−50.11 (−153.06–89.80)	43.69 (−102.27–126.91)	0.73
	sTNF-RI (pg/mL)	−25.08 (−209.51–186.66)	1.27 (−236.80–168.27)	0.92
	sTNF-RII (pg/mL)	−121.65 (−457.93–467.29)	99.33 (−484.31–477.72)	0.93
	D-dimer (ng/mL)	−149.05 (−294.46–58.72)	−45.85 (−295.57–142.43)	0.61
	oxLDL (U/L)	20,833.33 (2397.72–61,644.46)	8639.79 (−1716.35–43,693.14)	0.31
	IL−6 (pg/mL)	−0.47 (−0.85–0.61)	−0.02 (−0.88–0.72)	0.51
	VCAM (ng/mL)	66.69 (−2.65–187.38)	49.53 (−55.61–170.15)	0.38
	ICAM (ng/mL)	−2.03 (−28.37–22.30)	2.35 (−29.19–42.46)	0.45
	IP−10 (pg/mL)	0.23 (−44.28–29.08)	−9.99 (−31.65–20.27)	0.51
** Gut Integrity Markers **				
	LBP (μg/mL)	0.47(−4.8–8.35)	0.72 (−4.52–6.9)	0.94
	IFAB (pg/mL)	100.82 (−747.86–953.32)	58.89 (−1049.29–654.96)	0.46
	BDG (pg/mL)	−28.19 (−100.43–4.02)	8.36 (−63.26–76.12)	0.14
	Zonulin (μg/mL)	78.2 (−138–506.86)	270.54 (59.36–600.45)	0.15
	** Relative change (%) **			
** Zinc Level **				
	Zinc (μg/dL)	12.92 (2.36–31.15)	53.17 (19.44–87.08)	0.0001
** HIV Profile **				
	CD4+ T-cell count	0.00 (−14.12–9.81)	0.00 (−13.26–4.91)	0.80
** Metabolic Markers **				
	BMI (kg/m^2^)	0.78 (−2.30–3.23)	0.42 (−2.83–3.36)	0.68
	Weight (lbs)	1.36 (−2.13–4.25)	0.00 (−2.83–3.05)	0.46
	Waist-umbilicus (cm)	0.62 (−1.90–2.43)	1.67 (−2.76–3.49)	0.52
	Systolic blood pressure (mmHg)	4.65 (−3.10–9.85)	0.00 (−7.26–7.83)	0.59
	Diastolic blood pressure (mmHg)	3.49 (−6.41–7.23)	0.00 (−6.49–6.49)	0.90
	Non-HDL cholesterol (mg/dL)	3.97 (−16.95–30.09)	0.78 (−9.91–10.67)	0.60
	HDL (mg/dL)	−3.51 (−9.81–3.51)	−4.94 (−16.16–2.44)	0.42
	LDL (mg/dL)	9.23 (−25.23–25.40)	−1.95 (−18.27–14.05)	0.32
	VLDL (mg/dL)	3.33 (−22.47–22.05)	12.50 (−7.27–36.36)	0.23
	Cholesterol (mg/dL)	1.72 (−12.50–17.59)	−1.59 (−12.82–6.91)	0.73
	Chol:HDL ratio	3.45 (−7.41–17.39)	4.08 (−3.47–13.13)	0.64
	Triglycerides (mg/dL)	0.73 (−23.18–20.45)	10.66 (−8.42–36.46)	0.19
	10-year ASCVD (%)	10.00 (−15.94–25.51)	0.00 (−27.87–28.00)	0.91
** Endothelial Function **				
	Reactive Hyperemic Index	−18.54 (−31.73–10.74)	−10.16 (−27.27–13.79)	0.74
	Augmentation Index	0.00 (−69.70–75.00)	−7.14 (−52.94–50.00)	0.99
** Inflammatory Markers **				
	hsCRP (ng/mL)	−1.58 (−41.77–76.81)	1.10 (−33.16–79.09)	0.84
	sCD14 (ng/mL)	6.57 (−5.25–17.33)	−2.87 (−15.74–7.84)	0.02
	sCD163 (ng/mL)	−6.91 (−19.48–14.43)	7.63 (−17.77–28.19)	0.25
	sTNF-RI (pg/mL)	−2.19 (−22.55–17.72)	0.11 (−15.68–16.26)	0.71
	sTNF-RII (pg/mL)	−4.32 (−20.66–24.72)	5.48 (−21.53–19.40)	0.69
	D-dimer (ng/mL)	−22.00 (−54.24–12.08)	−16.75 (−45.32–32.14)	0.32
	oxLDL (U/L)	50.91 (4.50–171.86)	15.24 (−4.46–81.64)	0.20
	IL−6 (pg/mL)	−26.59 (−50.50–32.77)	−1.67 (−43.43–45.02)	0.29
	VCAM (ng/mL)	7.95 (−0.27–26.61)	6.44 (−4.73–21.81)	0.39
	ICAM (ng/mL)	−1.64 (−13.54–6.81)	1.52 (−10.99–17.82)	0.10
	IP−10 (pg/mL)	0.20 (−30.21–25.82)	−9.95 (−18.18–18.14)	0.56
** Gut Integrity Markers **				
	LBP (ng/mL)	3.56 (−28.23–55.41)	6.80 (−25.55–53.60)	0.89
	IFAB (pg/mL)	4.74 (−37.57–75.31)	3.95 (−39.73–52.54)	0.42
	BDG (pg/mL)	−25.89 (−51.03–3.45)	9.71 (−38.45–85.52)	0.13
	Zonulin (ng/mL)	15.70 (−14.77–56.66)	25.70 (10.18–84.41)	0.17

* The absolute changes were computed as (week 24–baseline), and the relative (percentage) changes were computed as ((week 24–baseline)/baseline) × 100. BMI = body mass index; LDL = low-density lipoprotein; HDL = high-density lipoprotein; VLDL = very-low-density lipoprotein; ASCVD: atherosclerotic cardiovascular disease; hsCRP = high-sensitivity C-reactive protein; sCD14 = soluble CD14; sCD163 = soluble CD163; sTNF-RI = soluble tumor necrosis factor receptor−1; sTNF-RII = soluble tumor necrosis factor receptor−2; oxLDL = oxidized LDL; IL−6 = interleukin−6; VCAM: vascular cell adhesion molecule−1; ICAM = intercellular adhesion molecule−1; IP-10 = interferon-gamma-inducible protein of 10 kDa; LBP = lipopolysaccharide-binding protein; IFAB = intestinal fatty-acid-binding protein; BDG = (1,3)-β-d-glucan.

**Table 3 nutrients-17-01671-t003:** Effects of zinc supplement in the treatment group compared to the placebo group.

	Outcome Variable	$\mathrm{Effects},\;\hat{\beta }$	SE ($\hat{\beta }$)	$95\%\;\mathrm{CI}\;\mathrm{of}\;\hat{\beta }$	*p*-Value
** Zinc Level **					
	Ln zinc (μg/dL)	0.71	0.14	0.43, 0.98	<0.01
** Metabolic Markers **					
	BMI (kg/m^2^)	−0.35	2.01	−4.29, 3.60	0.86
	Weight (lbs)	−2.99	12.23	−26.97, 20.99	0.81
	Ln waist-umbilicus (cm)	−0.01	0.03	−0.08, 0.05	0.74
	Ln systolic blood pressure (mmHg)	−0.02	0.03	−0.07, 0.03	0.41
	Diastolic blood pressure (mmHg)	−1.68	2.06	−5.71, 2.36	0.42
	Ln non-HDL cholesterol (mg/dL)	0.01	0.07	−0.12, 0.15	0.84
	Ln HDL (mg/dL)	−0.08	0.06	−0.19, 0.04	0.18
	Ln LDL (mg/dL)	−0.02	0.09	−0.19, 0.15	0.84
	Ln VLDL (mg/dL)	0.07	0.11	−0.15, 0.29	0.55
	Ln cholesterol (mg/dl)	−0.01	0.05	−0.11, 0.09	0.78
	Ln Chol:HDL ratio	0.06	0.07	−0.07, 0.20	0.36
	Ln triglycerides (mg/dL)	0.07	0.12	−0.17, 0.31	0.58
	Ln 10-year ASCVD (%)	−0.04	0.11	−0.26, 0.17	0.70
	Metabolic syndrome	−0.44	0.47	−1.36, 0.48	0.35
** Endothelial Function **					
	Reactive Hyperemic Index	−0.03	0.07	−0.17, 0.10	0.62
	Augmentation Index	0.59	2.43	−4.19, 5.36	0.81
** Inflammatory Markers **					
	Ln hsCRP (ng/mL)	−0.002	0.29	−0.57, 0.57	1.00
	Ln sCD14 (ng/mL)	−0.08	0.05	−0.18, 0.01	0.08
	Ln sCD163 (ng/mL)	−0.03	0.09	−0.21, 0.15	0.77
	Ln sTNF-RI (pg/mL)	0.04	0.06	−0.09, 0.16	0.55
	Ln sTNF-RII (pg/mL)	−0.004	0.07	−0.13, 0.13	0.95
	Ln D-dimer (ng/mL)	0.2	0.15	−0.10, 0.50	0.19
	oxLDL (U/L)	1819.18	8530.03	−14,899.37, 18,537.74	0.83
	Ln IL−6 (pg/mL)	0.07	0.19	−0.30, 0.43	0.73
	Ln VCAM (ng/mL)	−0.19	0.11	−0.41, 0.04	0.10
	ICAM (ng/mL)	0.41	20.19	−39.16, 39.98	0.98
	Ln IP−10 (pg/mL)	0.12	0.18	−0.22, 0.47	0.48
** Gut Integrity Markers **					
	Ln LBP (ng/mL)	0.05	0.14	−0.21, 0.32	0.69
	Ln IFAB (pg/mL)	0.02	0.14	−0.26, 0.30	0.90
	Ln BDG (pg/mL)	0.18	0.15	−0.12, 0.47	0.24
	Ln zonulin (ng/mL)	0.09	0.13	−0.16, 0.34	0.49

BMI = body mass index; LDL = low-density lipoprotein; HDL = high-density lipoprotein; VLDL = very-low-density lipoprotein; ASCVD: atherosclerotic cardiovascular disease; hsCRP = high-sensitivity C-reactive protein; sCD14 = soluble CD14; sCD163 = soluble CD163; sTNF-RI = soluble tumor necrosis factor receptor−1; sTNF-RII = soluble tumor necrosis factor receptor−2; oxLDL = oxidized LDL; IL−6 = interleukin−6; VCAM: vascular cell adhesion molecule−1; ICAM = intercellular adhesion molecule−1; IP-10 = interferon-gamma-inducible protein of 10 kDa; LBP = lipopolysaccharide-binding protein; IFAB = intestinal fatty-acid-binding protein; BDG = (1,3)-β-d-glucan.

**Table 4 nutrients-17-01671-t004:** Summary results for stratified analysis (stratified by median zinc level at baseline).

		Above-Median Zinc Group, Beta (*p*-Value)		Below-Median Zinc Group, Beta (*p*-Value)	
		N = 46		N = 47	
**Zinc Level**					
	Zinc (μg/dL)	23.8	0.04	32.02	0.001
**Metabolic Markers**					
	BMI (kg/m^2^)	−2.67	0.37	2.8	0.31
	Ln weight (lbs)	−0.2	0.12	0.17	0.15
	Ln waist-umbilicus (cm)	−0.06	0.24	0.04	0.35
	Ln systolic blood pressure (mmHg)	−0.05	0.18	−0.001	0.97
	Diastolic blood pressure (mmHg)	−5.67	0.06	1.48	0.60
	Ln non-HDL cholesterol (mg/dL)	−0.06	0.58	0.1	0.30
	Ln HDL (mg/dL)	0.02	0.85	−0.21	0.01
	Ln LDL (mg/dL)	−0.08	0.55	0.03	0.77
	Ln VLDL (mg/dL)	−0.01	0.97	0.21	0.23
	Ln cholesterol (mg/dL)	−0.04	0.58	0.01	0.90
	Chol:HDL ratio	0.26	0.72	0.71	0.25
	Triglycerides (mg/dL)	15.76	0.66	30.66	0.4
	10-year ASCVD (%)	−2.59	0.29	2.1	0.47
	Metabolic syndrome	−0.87	0.22	0.12	0.85
**Endothelial Function**					
	Reactive Hyperemic Index	−0.08	0.49	0.10	0.55
	Augmentation Index	4.95	0.30	−4.25	0.38
**Inflammatory Markers**					
	hsCRP (ng/mL)	−223.09	0.79	7.96	1.00
	Ln sCD14 (ng/mL)	−0.08	0.27	−0.12	0.07
	Ln sCD163 (ng/mL)	−0.06	0.63	−0.05	0.74
	Ln sTNF-RI (pg/mL)	0.07	0.41	0.02	0.83
	Ln sTNF-RII (pg/mL)	0.07	0.51	−0.03	0.75
	Ln D-dimer (ng/mL)	0.10	0.65	0.24	0.27
	Ln oxLDL (U/L)	0.01	0.96	−0.06	0.70
	Ln IL−6 (pg/mL)	0.08	0.75	0.07	0.81
	VCAM (ng/mL)	−84.53	0.48	−99.09	0.32
	ICAM (ng/mL)	22.08	0.31	−20.7	0.54
	Ln IP−10 (pg/mL)	0.2	0.44	0.11	0.62
**Gut Integrity Markers**					
	LBP (ng/mL)	−2.64	0.51	4.25	0.24
	Ln IFAB (pg/mL)	0.23	0.23	−0.21	0.32
	Ln BDG (pg/mL)	0.31	0.11	0.15	0.50
	Ln zonulin (ng/mL)	0.31	0.03	−0.13	0.53

BMI = body mass index; LDL = low-density lipoprotein; HDL = high-density lipoprotein; VLDL = very-low-density lipoprotein; ASCVD: atherosclerotic cardiovascular disease; hsCRP = high-sensitivity C-reactive protein; sCD14 = soluble CD14; sCD163 = soluble CD163; sTNF-RI = soluble tumor necrosis factor receptor−1; sTNF-RII = soluble tumor necrosis factor receptor−2; oxLDL = oxidized LDL; IL−6 = interleukin−6; VCAM: vascular cell adhesion molecule−1; ICAM = intercellular adhesion molecule−1; IP-10 = interferon-gamma-inducible protein of 10 kDa; LBP = lipopolysaccharide-binding protein; IFAB = intestinal fatty-acid-binding protein; BDG = (1,3)-β-d-glucan.

**Table 5 nutrients-17-01671-t005:** Summary results for correlation analysis (between zinc and other biomarkers).

	Other Biomarkers	At Baseline		At Week 24	
		Correlation Coefficient	*p*-Value	Correlation Coefficient	*p*-Value
** Metabolic Markers **					
	BMI (kg/m^2^)	0.0511	0.63	−0.0771	0.50
	Weight (lbs)	0.0863	0.41	−0.0844	0.46
	Waist-umbilicus (cm)	−0.0115	0.91	0.0191	0.87
	Systolic blood pressure (mmHg)	0.1029	0.33	0.1457	0.21
	Diastolic blood pressure (mmHg)	0.1896	0.07	0.1668	0.15
	Non-HDL cholesterol (mg/dL)	−0.0061	0.95	0.1146	0.32
	HDL (mg/dL)	0.1047	0.32	−0.1329	0.25
	LDL (mg/dL)	0.016	0.88	0.0113	0.92
	VLDL (mg/dl)	−0.0404	0.71	0.2529	0.03
	Cholesterol (mg/dL)	0.0392	0.71	0.0711	0.54
	Chol:HDL ratio	−0.0338	0.75	0.1935	0.09
	Triglycerides (mg/dL)	−0.0227	0.83	0.2715	0.02
	10-year ASCVD (%)	0.1096	0.29	0.286	0.01
	Metabolic syndrome	−0.0715	0.49	−0.0696	0.55
** Endothelial Function **					
	Reactive Hyperemic Index	0.0013	0.99	0.0572	0.64
	Augmentation Index	0.0335	0.75	0.1219	0.30
** Inflammatory Markers **					
	hsCRP (ng/mL)	0.068	0.52	−0.0551	0.65
	sCD14 (ng/mL)	−0.0835	0.43	−0.1329	0.27
	sCD163 (ng/mL)	−0.0687	0.51	0.0484	0.69
	sTNF-RI (pg/mL)	−0.0059	0.96	−0.0052	0.97
	sTNF-RII (pg/mL)	0.031	0.77	−0.0266	0.82
	D-dimer (ng/mL)	0.0908	0.39	0.0623	0.60
	oxLDL (U/L)	−0.0415	0.69	0.1365	0.25
	IL−6 (pg/mL)	−0.0262	0.80	−0.149	0.21
	VCAM (ng/mL)	−0.2043	0.05	0.0891	0.46
	ICAM (ng/mL)	−0.0278	0.79	−0.0419	0.73
	IP−10 (pg/mL)	0.0874	0.40	−0.0372	0.76
** Gut Integrity Markers **					
	LBP (ng/mL)	−0.0923	0.38	0.0987	0.41
	IFAB (pg/mL)	−0.2286	0.03	−0.0082	0.95
	BDG (pg/mL)	−0.0639	0.54	0.1719	0.15
	Zonulin (ng/mL)	0.0422	0.69	−0.0699	0.56

BMI = body mass index; LDL = low-density lipoprotein; HDL = high-density lipoprotein; VLDL = very-low-density lipoprotein; ASCVD: atherosclerotic cardiovascular disease; hsCRP = high-sensitivity C-reactive protein; sCD14 = soluble CD14; sCD163 = soluble CD163; sTNF-RI = soluble tumor necrosis factor receptor−1; sTNF-RII = soluble tumor necrosis factor receptor−2; oxLDL = oxidized LDL; IL−6 = interleukin−6; VCAM: vascular cell adhesion molecule−1; ICAM = intercellular adhesion molecule−1; IP-10 = interferon-gamma-inducible protein of 10 kDa; LBP = lipopolysaccharide-binding protein; IFAB = intestinal fatty-acid-binding protein; BDG = (1,3)-β-d-glucan.

## Data Availability

The original contributions presented in this study are included in the article. Further inquiries can be directed to the corresponding author.

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
