# Peer review of "Zinc Supplementation, Inflammation, and Gut Integrity Markers in HIV Infection: A Randomized Placebo-Controlled Trial"

_nutrients, 2025, doi:10.3390/nu17101671_

Round 1
Reviewer 1 Report
Comments and Suggestions for Authors
This is a very interesting study. It is scientifically sound and beautifully written. It would be interesting to see if there were any differences due to the types of antiviral therapy the patients were receiving. If you have that information, I would include it
Author Response
Thank you for your review and comment.
That is a great idea, but the majority of the participant in this study are on integrase inhibitors, therefore, we cannot look into this in the current study.
However, we will include in result section the following: ‘Across both treatment arms, 79% were on integrase inhibitors, 15% on protease inhibitors (PI), and 20% on non-nucleoside reverse transcriptase inhibitors (NNRTIs).”
Reviewer 2 Report
Comments and Suggestions for Authors
The article makes several original and field-specific contributions. Here's a summary of its relevance and the gap it addresses. It is the first placebo-controlled clinical trial to investigate zinc supplementation in virologically suppressed people living with HIV (PLHIV), specifically focusing on inflammation, cardiovascular, and gut integrity markers. Based on the article the following specific methodological improvements and additional controls are suggested.
- The 24-week intervention may be too short to observe significant changes in inflammation and gut integrity markers. A longer duration may reveal delayed or cumulative effects.
- The study did not measure or control for dietary zinc intake. Including dietary assessments would help isolate the supplementation effect from dietary variability.
- Although major inflammatory conditions were exclusionary, subclinical inflammation from obesity, smoking, or metabolic syndrome may have influenced outcomes. Stratified or sensitivity analyses adjusting for these confounders would strengthen conclusions.
Implementing these improvements and controls would enhance the study's rigor, validity, and policy implications.
Author Response
Thank you for your valuable comments.
- A longer duration of treatment could be considered in future studies.
- While dietary zinc intake was not directly measured, participants were randomly assigned to the treatment and control groups, which should help balance baseline dietary zinc intake between groups and mitigate its potential confounding effect. However, this should be included in future studies
- Although subclinical inflammation from factors such as obesity, smoking, or metabolic syndrome may influence inflammatory status, baseline measurements of inflammatory markers showed no statistically significant differences between the treatment and control groups. This suggests that randomization was effective in balancing potential confounders related to subclinical inflammation. Nonetheless, we acknowledge the value of stratified or sensitivity analyses and will consider these approaches in future studies.
Reviewer 3 Report
Comments and Suggestions for Authors
Excellent article.
Comments on the Quality of English LanguageExcellent article.
Author Response
Thank you for your review and comment.